# Functional expression of the nitrogenase Fe protein in transgenic rice

Can Baysal [1,5,6], Stefan Burén [2,3,6], Wenshu He[1], Xi Jiang [2,3], Teresa Capell[1], Luis M. Rubio [2,3✉] & Paul Christou [1,4✉]

Engineering cereals to express functional nitrogenase is a long-term goal of plant bio-technology and would permit partial or total replacement of synthetic N fertilizers by metabolization of atmospheric $N_2$. Developing this technology is hindered by the genetic and biochemical complexity of nitrogenase biosynthesis. Nitrogenase and many of the accessory proteins involved in its assembly and function are $O_2$ sensitive and only sparingly soluble in non-native hosts. We generated transgenic rice plants expressing the nitrogenase structural component, Fe protein (NifH), which carries a [4Fe-4S] cluster in its active form. NifH from *Hydrogenobacter thermophilus* was targeted to mitochondria together with the putative peptidyl prolyl *cis-trans* isomerase NifM from *Azotobacter vinelandii* to assist in NifH polypeptide folding. The isolated NifH was partially active in electron transfer to the MoFe protein nitrogenase component (NifDK) and in the biosynthesis of the nitrogenase iron-molybdenum cofactor (FeMo-co), two fundamental roles for NifH in $N_2$ fixation. NifH functionality was, however, limited by poor [4Fe-4S] cluster occupancy, highlighting the importance of in vivo [Fe-S] cluster insertion and stability to achieve biological $N_2$ fixation *in planta*. Nevertheless, the expression and activity of a nitrogenase component in rice plants represents the first major step to engineer functional nitrogenase in cereal crops.

[1] Department of Plant Production and Forestry Science, University of Lleida-Agrotecnio CERCA Center, Av. Alcalde Rovira Roure, 191, 25198 Lleida, Spain. [2] Centro de Biotecnología y Genómica de Plantas, Universidad Politécnica de Madrid (UPM) - Instituto Nacional de Investigación y Tecnología Agraria y Alimentaria (INIA), Campus Montegancedo UPM, 28223 Pozuelo de Alarcón (Madrid), Spain. [3] Departamento de Biotecnología-Biología Vegetal, Escuela Técnica Superior de Ingeniería Agronómica, Alimentaria y de Biosistemas, Universidad Politécnica de Madrid, 28040 Madrid, Spain. [4] ICREA, Catalan Institute for Research and Advanced Studies, Passeig Lluís Companys 23, 08010 Barcelona, Spain. [5] Present address: Department of Genetics, Cell Biology and Development, University of Minnesota, St. Paul, MN, USA. [6] These authors contributed equally: Can Baysal, Stefan Burén. ✉email: lm.rubio@upm.es; paul.christou@udl.cat

Crops utilize nitrogen (N) mainly in two different forms: nitrate ($NO_3^-$) and ammonium ($NH_4^+$). N is one of the major components of chlorophyll, energy-transfer compounds such as ATP, nucleic acids, and proteins. Therefore, N availability in crops controls the rate of photosynthesis, cell growth, metabolism, and protein synthesis[1]. Crops are dependent on an adequate N supply, typically obtained from industrial synthetic fertilizers, but they do not assimilate more than half of the N applied as fertilizer, the remainder spilling over or leaching from the soil as a major source of pollution[2–4]. It is therefore important to explore strategies that will reduce the dependence of agriculture on N fertilizers.

Biological nitrogen fixation (BNF) is the reduction of $N_2$ gas to ammonia ($NH_3$) by the enzyme nitrogenase[5]. BNF is widespread in prokaryotes (bacteria and archaea), but no eukaryote species is yet known to directly convert $N_2$ into a biologically useful form[6]. The direct transfer of nitrogenase genes from prokaryotes to crops is one of the most ambitious strategies to achieve BNF in plants[7,8]. Nitrogenases are composed of two $O_2$-sensitive interacting metalloproteins: dinitrogenase and dinitrogenase reductase[5]. In the molybdenum (Mo) nitrogenase, which is the most abundant and best characterized form, these components are known as the MoFe protein (encoded by *nifD* and *nifK*) and Fe protein (encoded by *nifH*), respectively. In addition, multiple accessory components are required for nitrogenase assembly and activity[9]. The MoFe protein ($NifD_2K_2$) includes two P-clusters [8Fe-7S] and two FeMo-cofactors [7Fe-9S-C-Mo-*R* homocitrate], and its role is to bind and reduce $N_2$. The Fe protein is a NifH homodimer that presents a single [4Fe-4S] cluster linking the two subunits and one site for MgATP binding and hydrolysis in each subunit[10]. The Fe protein is the obligate electron donor to the MoFe protein for $N_2$ reduction[11]. It is also required for the maturation and activation of the MoFe protein P-cluster and FeMo-co[9]. Because the Fe protein is more sensitive to $O_2$ than the MoFe protein, and because it is also required for the biosynthesis of MoFe protein cofactors, its functional expression has been used as proof of principle for nitrogenase engineering in eukaryotes[12].

Mitochondria and chloroplasts are energy-conversion organelles in eukaryotic cells with the capacity to provide low potential electrons and ATP needed for BNF. The low-$O_2$ environment of mitochondria also protects nitrogenase from $O_2$ damage[12]. Both organelles have therefore been tested for the expression of nitrogenase Fe protein[12–15]. Active Fe protein was produced by expressing NifH and NifM as the minimal complement, optionally together with NifS and NifU depending on [Fe-S] cluster biosynthesis and insertion by the host species. NifM is a putative peptidyl prolyl *cis-trans* isomerase that facilitates Fe protein folding and improves its solubility[16,17]. NifS mobilizes S from cysteine for the synthesis of [Fe-S] clusters on NifU, which subsequently donates them to cluster-less apo-Fe protein[18,19]. When NifH and NifM were co-expressed in the mitochondria of the yeast *Saccharomyces cerevisiae*, NifU and NifS were not required because the mitochondrial [Fe-S] cluster biosynthetic machinery was able to load [4Fe-4S] clusters onto the apo-NifH protein efficiently[12]. However, when NifH was co-expressed transiently with NifM, NifS and NifU in the mitochondria of *Nicotiana benthamiana*, the nitrogenase activity was lower than that achieved in yeast, suggesting that success of [4Fe-4S] cluster insertion ultimately depends on the [Fe-S] cluster machinery of the host[15].

Here, we engineered rice to express selected *nifH* and *nifM* genes, targeting the corresponding proteins to the mitochondrial matrix to minimize $O_2$ damage and produce correctly folded and enzymatically active NifH. Soluble NifH polypeptides accumulated in rice mitochondria and the as-isolated Fe protein showed limited functionality in electron donation to the MoFe protein and in FeMo-co synthesis, two fundamental activities to engineer

nitrogenase. In vitro transfer of [4Fe-4S] clusters from the NifU donor to NifH was necessary to achieve maximum Fe protein activity. This result indicates that even though NifH incorporated some endogenous rice mitochondrial [4Fe-4S] clusters leading to activity, much of it accumulated as apo-protein, thereby identifying [Fe-S] cluster biosynthesis, insertion and stability as areas that should be the focus of future research.

## Results

**Transformation and recovery of transgenic rice callus and plants expressing NifH and NifM targeted to the mitochondria.** The genes encoding *Hydrogenobacter thermophilus* NifH (NifH$^{Ht}$) and *Azotobacter vinelandii* NifM (NifM$^{Av}$) were cloned in separate vectors and introduced into rice along with the hygromycin phosphotransferase (*hpt*) gene for selection. The *nifH$^{Ht}$* and *nifM$^{Av}$* sequences were previously codon optimized for *S. cerevisiae* and expressed in both *S. cerevisiae* and *N. benthamiana*[15]. The same codon-optimized *nifH$^{Ht}$* and *nifM$^{Av}$* sequences were used because no rare codons were present in the two genes in the context of rice codon usage[20]. Expression of the *nifH$^{Ht}$* gene was driven by the strong constitutive maize ubiquitin promoter and first intron (*ZmUbi1* + 1sti). The N-terminal 29-amino-acid mitochondrial transit peptide from *S. cerevisiae* cytochrome *c* oxidase subunit IV (Cox4) was added to the coding sequence, based on previous work showing its ability to direct GFP to rice mitochondria[21]. The 28-amino-acid Twinstrep (TS) tag was also added to facilitate NifH$^{Ht}$ detection and purification. This tag did not affect NifH activity in *A. vinelandii*, *S. cerevisiae* or *N. benthamiana*[14,15]. An auxiliary DNA vector was constructed to co-express *nifM$^{Av}$* under the control of the constitutive *OsActin* promoter. In this case, an N-terminal transit peptide from *Neurospora crassa* subunit 9 (Su9), was used because it was shown to deliver GFP to rice mitochondria[21] and NifM to *N. benthamiana* mitochondria[15]. The process for the recovery of transgenic rice callus expressing *nif* transgenes and the regeneration of transgenic plants is shown in Supplementary Fig. 1.

**Co-expression of *Os*NifH$^{Ht}$ and *Os*NifM$^{Av}$.** The expression of *Os*NifH$^{Ht}$ and *Os*NifM$^{Av}$ was analyzed at the mRNA and protein levels in three independent rice callus lines and the corresponding regenerated plants (Fig. 1 and Supplementary Fig. 2). Both genes were expressed at higher levels in plants than in the callus (Fig. 1a, b). The *nifH$^{Ht}$* gene was placed under a stronger promoter than *nifM$^{Av}$* because the former encodes the most abundant Nif protein and it is part of the nitrogenase complex. Although there is no study proving that NifM is essential for NifH in plant mitochondria, NifM has been shown to be required for NifH solubility in heterologous expression systems including *E. coli*, yeast mitochondria, and plant chloroplasts[14,16,22]. Although we were unable to obtain clear western blot signals for *Os*NifM$^{Av}$ in the plant extracts, the fact that *Os*NifH$^{Ht}$ was soluble indicated that *Os*NifM$^{Av}$ was expressed at sufficient levels to facilitate NifH polypeptide maturation, as recently shown for NifH$^{Ht}$ in *S. cerevisiae*[22]. In callus, both *Os*NifH$^{Ht}$ and *Os*NifM$^{Av}$ gave rise to bands of the expected size (~33 kDa for both proteins) when analyzed by SDS-PAGE and western blot, indicating that the proteins were correctly processed and stable in rice mitochondria (Fig. 1c, d). Analysis of *Os*NifH$^{Ht}$ protein expression in the T1 generation confirmed that *Os*NifH$^{Ht}$ was stably inherited and expressed in progeny (Fig. 1e, f).

**Purification of *Os*NifH$^{Ht}$.** *Os*NifH$^{Ht}$ was purified from rice callus and the corresponding regenerated plants by strep-tag affinity chromatography (STAC) (Fig. 2a, b; Supplementary Fig. 3). To

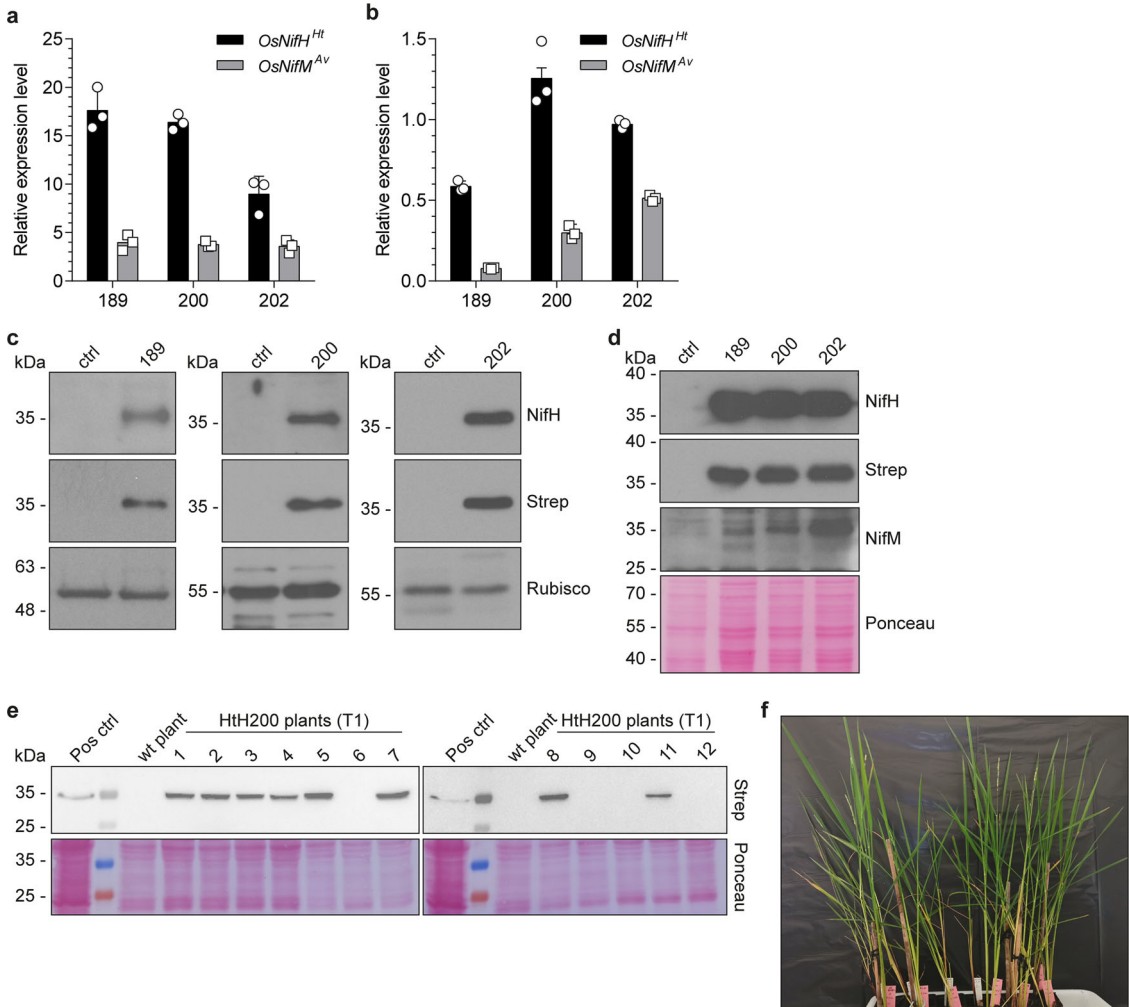

**Fig. 1 Expression of rice-derived *Os*NifH^Ht and *Os*NifM^Av.** Relative mRNA expression levels of *OsnifH^Ht* and *OsnifM^Av* in three different (independent biological replicates) rice plant lines (**a**) and the corresponding callus lines (**b**). Data (normalized to *OsActin* mRNA) are means ± SD (*n* = 3 technical replicates). Immunoblot analysis of soluble protein extracts from rice leaves (**c**) and callus (**d**) probed with antibodies against NifH, NifM, and the Strep-tag. Antibodies against RuBisCO were used as loading control for plant lines. Ponceau staining was used as loading control for callus extracts due to the low expression of RuBisCO. Ctrl lane shows non-transformed callus and plant lines. **e** Stable expression of *Os*NifH^Ht in the T1 segregating generation of rice plant line HtH200. Protein extract from callus expressing *Os*NifH^Ht (line HtH206) was used as positive control (Pos ctrl). Uncropped immunoblots are shown in Supplementary Figs. 6–10. **f** Phenotype of *Os*NifH^Ht expressing T1 progeny showing normal growth and development.

minimize O$_2$ exposure produced during photosynthesis and maximize the isolation of functional *Os*NifH^Ht, plants were Fe-fertilized and harvested before onset of light at the end of the dark period, following a procedure previously shown to be successful to obtain active NifH^Av and NifH^Ht from tobacco chloroplasts and mitochondria[15,23]. Purified *Os*NifH^Ht was mostly soluble and could be isolated from plant line Ht200 with yields of 0.5 mg kg$^{-1}$ fresh weight (1-month-old plants) or 0.25 mg kg$^{-1}$ fresh weight (2-month-old plants). The yield of *Os*NifH^Ht from three independent callus lines was ~2.5 mg kg$^{-1}$ (line Ht189), ~7 mg kg$^{-1}$ (line Ht200) and ~6.5 mg kg$^{-1}$ (line Ht202) (Supplementary Fig. 4). Purified *Os*NifH^Ht from callus and plants migrated as a single major band indicating correct processing (Fig. 2a, b). The side-by-side comparison of *Sc*NifH^Ht and *Os*NifH^Ht (both targeted to the mitochondria) was also consistent with the correct processing of *Os*NifH^Ht in the rice mitochondrial matrix, resulting in proteins with molecular weights of ~33 kDa (Fig. 2c). N-terminal sequencing of purified *Os*NifH^Ht revealed a major product with the anticipated N-terminal residues QKP following the cleavage of the Cox4 peptide, together with three minor variants representing alternative processing events (Fig. 2d). Mass

spectrometry identified the *Os*NifH^Ht protein with 63% sequence coverage and confirmed that the C-terminus of the protein was intact (Supplementary Fig. 5).

**Purified *Os*NifH^Ht shows Fe protein activity.** The Fe protein activity of purified *Os*NifH^Ht was determined in vitro using an acetylene reduction assay after mixing with NifDK purified from *A. vinelandii* (NifDK^Av). The *Os*NifH^Ht protein isolated from rice leaves showed low activity (~7 nmol ethylene min$^{-1}$ mg$^{-1}$ NifDK^Av) (Fig. 3a), probably limited by the relatively low yield of NifH in the plant tissue and hence the low amount of NifH (~8.9 μg) and low ratio of NifH:NifDK (~10:1) in the assay. To confirm the functionality of the as-isolated rice Fe protein, we performed titration experiments using increasing amounts of *Os*NifH^Ht purified from three independent callus lines. As anticipated for a functional Fe protein, higher activity was observed with an increasing NifH:NifDK ratio (Fig. 3b). Importantly, the activity of the *Os*NifH^Ht protein isolated from rice leaves was about half that of the callus protein at a similar ratio, which may reflect the use of more diluted plant *Os*NifH^Ht protein

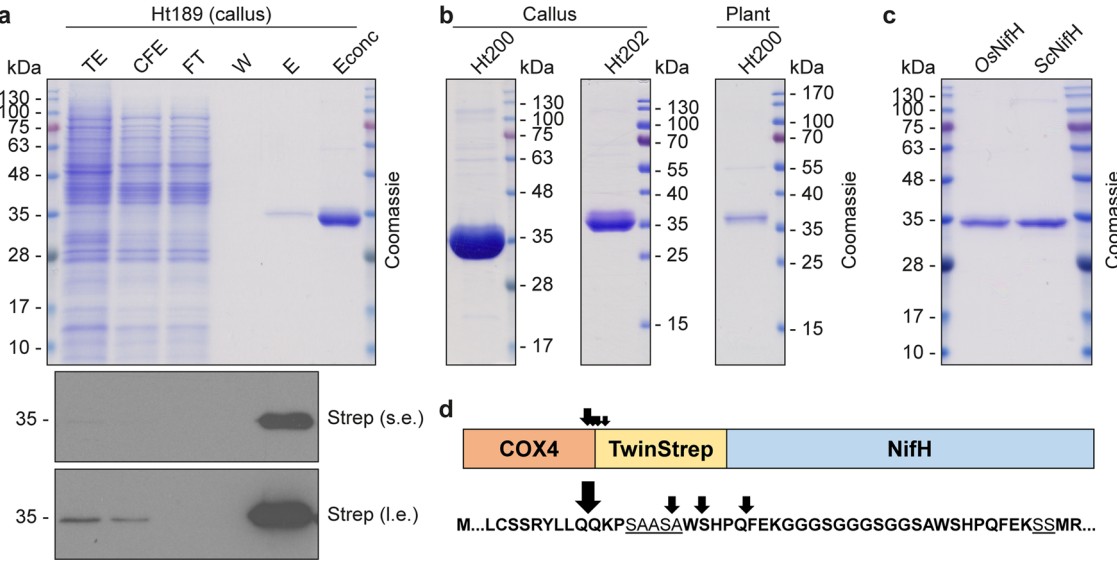

**Fig. 2 STAC purification of _Os_NifH^Ht protein targeted to rice mitochondria. a** Purification of _Os_NifH^Ht from Ht189 callus. Fractions were analyzed by SDS-PAGE followed by Coomassie staining or immunoblotting. TE total extract, CFE soluble cell-free extract, FT flow-through fraction, W wash fraction, E elution fraction. Econc concentrated elution fraction (final sample collected). s.e. short exposure, l.e. long exposure. Uncropped immunoblot analysis for the purification processes are shown in Supplementary Fig. 11. **b** Final _Os_NifH^Ht sample isolated from Ht200 and Ht202 callus lines, and Ht200 rice plants. Uncropped Coomassie gels and immunoblot analysis for the purification processes are shown in Supplementary Fig. 3. **c** Side-by-side comparison of NifH^Ht migration when isolated from Ht189 callus (_Os_NifH^Ht) or yeast (_Sc_NifH^Ht). **d** N-terminal sequencing of _Os_NifH^Ht isolated from Ht189 callus. The deduced Cox4 mitochondrial targeting peptide processing sites are indicated with arrows (the dominant processing size indicated by a larger arrow). The underlined sequences indicate peptide linkers between the Cox4 signal and the TS-tag (SAASA), and between the TS-tag and _Os_NifH^Ht (SS).

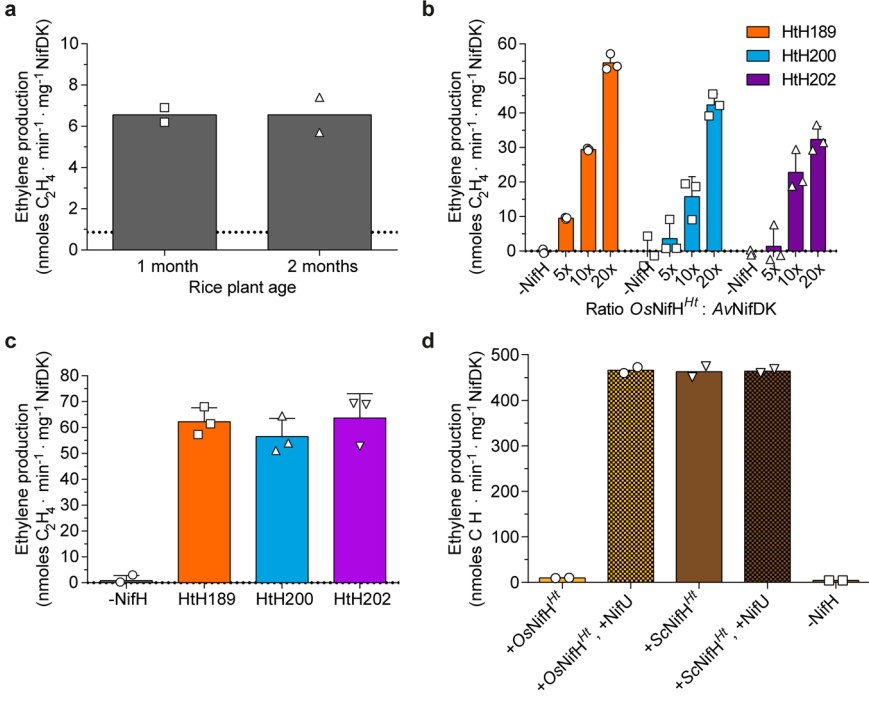

**Fig. 3 Acetylene reduction assay (ARA) with purified _Os_NifH^Ht. a** ARA using _Os_NifH^Ht isolated from HtH200 rice plants (10:1 ratio of _Os_NifH^Ht:NifDK^Av) (data are means ± SD, $n = 2$ technical replicates). Dotted line indicates the negative control ARA in the absence of NifH (mean of $n = 3$ technical replicates). The positive control ARA (40:1 ratio of NifH^Av:NifDK^Av) generated 1285 ± 225 units (mean ± SD, $n = 3$ technical replicates). **b** ARA with increasing amounts of _Os_NifH^Ht isolated from three different rice callus lines (0, 5:1, 10:1 and 20:1 ratio of _Os_NifH^Ht:NifDK^Av). The positive control ARA (20:1 ratio of NifH^Av:NifDK^Av) generated 1426 ± 43 units for Ht189, 1127 ± 65 units for Ht200 and 1074 ± 153 units for Ht202 (data are means ± SD, $n = 3$ technical replicates). **c** ARA using _Os_NifH^Ht isolated from three different rice callus lines at a 40:1 ratio of _Os_NifH^Ht:NifDK^Av. The positive control ARA (40:1 ratio of NifH^Av:NifDK^Av) generated 1285 ± 225 units (data are means ± SD, $n = 3$ technical replicates). **d** ARA using _Os_NifH^Ht and _Sc_NifH^Ht before and after reconstitution with [4Fe-4S] cluster-loaded _Ec_NifU^Av. The positive control ARA (using NifH^Av) generated 1553 ± 82 units. All reactions were performed with a 20:1 ratio of NifH:NifDK^Av (data are means ± SD, $n = 2$ technical replicates). All activities are reported as nmol ethylene formed per min and mg of NifDK^Av.

in the assay (we typically observed slight inhibition caused by large amounts of buffer).

$Os$NifH$^{Ht}$ appeared completely soluble in all three callus lines (Fig. 2a, Supplementary Fig. 3a, b), despite the varying expression level of NifM (Fig. 1b, d). The activities of these three $Os$NifH$^{Ht}$ proteins in the same experiment at the same NifH:NifDK ratio of 40:1 were almost identical (Fig. 3c). This indicated that $Os$NifM$^{Av}$ expression was sufficient in all three rice lines, which was consistent with recent results obtained in $S.$ $cerevisiae$[22].

We then supplemented the reaction mixture with pure $A.$ $vinelandii$ NifU expressed in $Escherichia$ $coli$ and loaded with [4Fe-4S] clusters ($Ec$NifU$^{Av}$). NifU is known to serve as [4Fe-4S] donor to apo-NifH in vivo[24] and in vitro[19]. Indeed, the reconstitution of $Os$NifH$^{Ht}$ by the in vitro transfer of [4Fe-4S] from NifU increased its activity ~9-fold (compare activities in Fig. 3c, d). Importantly, the activity of $Os$NifH$^{Ht}$ and $Sc$NifH$^{Ht}$ proteins was identical after reconstitution (Fig. 3d), suggesting that $Os$NifH$^{Ht}$ was correctly folded but not fully mature, probably due to insufficient insertion of the [4Fe-4S] cluster into $Os$NifH$^{Ht}$ in the rice mitochondria, or perhaps due to instability of the clusters as previously suggested in $N.$ $benthamiana$[15]. These results demonstrate that the activities of reconstituted rice and yeast Fe protein are comparable.

**Purified $Os$NifH$^{Ht}$ supports in vitro FeMo-co synthesis.** The isolated $Os$NifH$^{Ht}$ also supported in vitro FeMo-co synthesis, which is another fundamental role of NifH required for BNF. FeMo-co synthesis and the activation of apo-NifDK$^{Av}$ (containing P-clusters but devoid of FeMo-co) was measured in vitro by combining $Methanothermobacter$ $thermautotrophicus$ NifB expressed in yeast ($Sc$NifB$^{Mt}$), $Ec$NifU$^{Av}$ (provider of [4Fe-4S] clusters for NifB-co biosynthesis), $S$-adenosyl methionine (SAM), molybdate, homocitrate, apo-NifEN$^{Av}$ (isolated from a $\Delta nifB$ strain containing permanent [4Fe-4S] clusters but lacking the FeMo-co precursor), purified $Os$NifH$^{Ht}$ and apo-NifDK$^{Av}$.

The in vitro FeMo-co synthesis assay can be divided in three steps[9]. In the first step, NifB-co synthesis is catalyzed by NifB, which carries three [4Fe-4S] clusters that are provided by NifU[25,26]. This SAM-radical enzyme fuses two of its [4Fe-4S] clusters, incorporates an additional sulfide, and generates a carbide atom that is inserted at the center of this Fe-S frame. The result is an [8Fe-9S-C] called NifB-co that is then transferred to the NifEN scaffold to serve as FeMo-co precursor. In this step, NifU [4Fe-4S] clusters used as substrates for NifB are also likely to activate some of the cluster-less $Os$NifH$^{Ht}$ protein. In the second step, FeMo-co maturation occurs at NifEN upon incorporation of Mo and homocitrate, which depend on transient NifH-NifEN associations[27–30]. The de novo synthesized FeMo-co is transferred to apo-NifDK, converting the inactive protein into active holo-NifDK. In the third step, ARA is often used to assess the level of holo-NifDK activity (and hence FeMo-co synthesis).

Following the FeMo-co synthesis and insertion reactions, tetrathiomolybdate was added to prevent further FeMo-co incorporation into apo-NifDK$^{Av}$ during the acetylene reduction assay. $Os$NifH$^{Ht}$ and $Sc$NifB$^{Mt}$ jointly catalyzed the NifB-dependent synthesis of FeMo-co in vitro. FeMo-co synthesis confirmed the compatibility of $Os$NifH$^{Ht}$ and $Sc$NifB$^{Mt}$ as demonstrated by the in vitro maturation of NifDK (Fig. 4). It also demonstrated the interspecies compatibility of $Os$NifH$^{Ht}$, $Sc$NifB$^{Mt}$, NifEN$^{Av}$ and NifDK$^{Av}$, collectively establishing the conserved biochemical core of nitrogenase.

## Discussion

Engineering BNF in plants is a major longstanding goal of plant biotechnology. Earlier strategies to reduce global dependence on

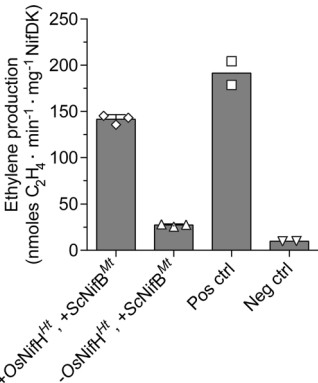

**Fig. 4 NifB-dependent in vitro FeMo-co synthesis using $Os$NifH$^{Ht}$ and $Sc$NifB$^{Mt}$.** In vitro FeMo-co synthesis was performed by combining $M.$ $thermautotrophicus$ NifB purified from yeast ($Sc$NifB$^{Mt}$) with $Os$NifH$^{Ht}$ (bar 1). The subsequent ARA was performed using a 20:1 molar ratio of NifH$^{Av}$ to NifDK$^{Av}$. To inhibit apo-NifDK$^{Av}$ activation by FeMo-co produced during the ARA, tetrathiomolybdate was added to all vials following in vitro FeMo-co synthesis and insertion into apo-NifDK$^{Av}$. Bar 2 represents the background activity observed from using $Sc$NifB$^{Mt}$ in the absence of $Os$NifH$^{Ht}$. As control reactions for the functionality of apo-NifEN$^{Av}$ and apo-NifDK$^{Av}$, FeMo-co synthesis was performed using $Sc$NifH$^{Ht}$ in the presence (Pos ctrl) or absence (Neg ctrl) of purified NifB-co. Data are means ± SD, $n = 3$ (bars 1 and 2) and $n = 2$ (bars 3 and 4) technical replicates.

N fertilizers included the use of diazotrophic bacteria to colonize the rhizosphere[31–36]. The introduction of bacterial $nif$ genes into cereals to increase crop productivity offers a more direct approach in which the plants fix their own N. Multigene transfer technology allows the optimization of different combinations of heterologous genes from diverse origins, as well regulatory elements such as promoters and targeting peptides. However, remaining challenges include the $O_2$ sensitivity of nitrogenase and its accessory proteins, the complexity of the machinery that provides metal clusters to nitrogenase components[9], and the intricate regulation of nitrogenase expression and activity[37].

Many $nif$ genes are involved in the assembly and activity of nitrogenase and its metal cofactors in bacteria. The essential bacterial genes that must be transferred to cereals because there are no corresponding endogenous proteins include at least $nifH$, $nifD$, $nifK$ $nifB$, $nifE$ and $nifN$. In contrast, some or all of the accessory genes, such as $nifU$, $nifS$, $nifQ$ and $nifV$ (which provide the FeMo-co building blocks: [Fe-S] clusters, molybdenum, and homocitrate), as well as $nifJ$ and $nifF$ (which are involved in electron transfer to NifH) show at least partial functional redundancy with plant proteins[7,38]. Plant counterparts of the NifU/NifS system are found in the mitochondria, which could therefore provide a ready source of [Fe-S] clusters as well as energy and the near anoxic environment necessary to protect the heterologous nitrogenase[39]. It is not yet possible to directly transform the plant mitochondrial genome, but proteins encoded by nuclear transgenes can be directed to the mitochondria if provided with the appropriate N-terminal targeting peptide.

The expression of an active Fe protein in plant mitochondria thus requires at least $nifH$ and $nifM$, and possibly also $nifU$ and $nifS$. When $Klebsiella$ $pneumoniae$ NifH was produced in $E.$ $coli$, only NifM was required for NifH functionality[16,40], suggesting that [4Fe-4S] cluster insertion is less specific and can be achieved by the host machinery to some extent. In yeast, NifU and NifS were needed for the provision of [Fe-S] clusters to confer NifB activity[41], but not for NifH[12]. This may reflect the distinct mechanisms used to incorporate [Fe-S] clusters into the NifB and NifH proteins, or the different requirements for the clusters.

While NifH contains a single permanent [4Fe-4S] cluster only required for catalysis, NifB has three distinct [4Fe-4S] clusters[9]. One is a SAM-coordinated cluster (the RS-cluster) that is needed for the catalytic activity of NifB, and the other two are [4Fe-4S] clusters (K1 and K2-clusters) that are the substrate clusters for NifB-co formation. The expression of active NifB therefore requires constant biosynthesis and insertion of [4Fe-4S] clusters to replenish those used for NifB-co. The heterologous host also influences the activity of Nif proteins. For example, when NifH was transiently co-expressed with NifM, NifU and NifS in *N. benthamiana* mitochondria, its activity was lower than in yeast[15]. In vitro reconstitution restored activity to *N. benthamiana* NifH, suggesting that the protein isolated from leaf mitochondria was to large extent lacking the required [4Fe-4S] clusters.

The first active Fe protein reported in higher plants was *A. vinelandii* NifH expressed together with NifM in transplastomic tobacco[13]. However, active Fe protein was only detected when the plants were incubated in a 10% $O_2$ atmosphere[13]. The same transplastomic plants were later grown in normal air, and Fe protein activity was detected when leaves were collected at the end of the dark period and when the protein extract was heated[23]. The heat treatment precipitated apo-NifH protein lacking [4Fe-4S] clusters, and therefore enriched the holo-protein, increasing the specific activity of the remaining NifH protein. The study suggested that the endogenous [Fe-S] cluster biosynthesis system in plastids was, like that in *N. benthamiana* mitochondria, insufficient for complete NifH maturation[23]. Similar results were reported by the transient expression of soluble *A. vinelandii* NifH, NifM, NifU and NifS in *N. benthamiana* chloroplasts[14]. NifH$^{Av}$ was active when the leaves were harvested at the end of the dark period, but only when co-expressed with NifU$^{Av}$ and NifS$^{Av}$, confirming that chloroplast assembly and transfer factors are unable to provide a sufficient quantity of [Fe-S] clusters for the assembly of a functional NifH[14].

After screening 32 diverse NifH proteins in *N. benthamiana* to compare expression level, solubility, and functionality NifH$^{Ht}$ was found to have superior properties for plant expression[15]. For this reason, here we used the TS-tag and STAC to purify NifH$^{Ht}$ from stably transformed rice plants and callus. The *Os*NifH$^{Ht}$ protein migrated as one major band in both rice callus and plants, confirming correct processing in the mitochondria. The isolated *Os*NifH$^{Ht}$ protein was colorless, but this was expected given its low concentration in rice extracts compared to those from bacteria or yeast. Most importantly, the *Os*NifH$^{Ht}$ protein was soluble and stable in rice mitochondria, although not fully equipped with its [4Fe-4S] cluster, and could readily be activated when complemented with [4Fe-4S] cluster-loaded NifU in vitro. This is essential because the Fe protein must be abundant, stable and soluble for successful nitrogenase engineering in plants. The accumulation of an unstable NifH protein could induce cellular stress and damage if exposed to $O_2$, or if [4Fe-4S] clusters were limited for other reasons. The efficient in vitro reconstitution of *Os*NifH$^{Ht}$ protein indicated that future research should focus on better Fe transport, [Fe-S] cluster biosynthesis, delivery to Nif proteins and protection. In this regard, repeated attempts to co-express NifU and NifS from *A. vinelandii* were unsuccessful in generating detectable protein accumulation in rice callus and regenerated plants despite high levels of mRNA expression (Supplementary Fig. 12). Alternatively, [Fe-S] cluster delivery to Nif proteins could be enhanced by manipulating the host Fe homeostasis (using Fe fortified rice lines) and endogenous mitochondrial pathways.

Our results agree with data previously reported for *N. benthamiana* leaves (transient expression). The yield of *Nb*NifH$^{Ht}$ was ~5–6 mg kg$^{-1}$ leaf tissue using STAC purification, but the activity of the as-isolated enzyme was low (maximum 25 nmol ethylene min$^{-1}$ mg$^{-1}$ when NifH was used at a 40-fold molar ratio to NifDK). However, NifH reconstituted by the in vitro transfer of [4Fe-4S] clusters from NifU increased the activity of *Nb*NifH$^{Ht}$ to 250 nmol ethylene min$^{-1}$ mg$^{-1}$. Although we purified relatively low amounts of NifH$^{Ht}$ from rice plants (0.25–0.5 mg kg$^{-1}$) compared to *N. benthamiana*, the yield of rice callus (~6 mg kg$^{-1}$) was similar to that reported in *N. benthamiana* and the activity of the protein from rice callus was similar to that of the protein from *N. benthamiana* leaves. Furthermore, the activity of reconstituted *Os*NifH$^{Ht}$ was ~9-fold higher than the as-isolated enzyme (50 vs 450 nmol ethylene min$^{-1}$ mg$^{-1}$) and identical to NifH$^{Ht}$ expressed in yeast, which probably reflects its maximum activity in this assay with a full complement of [4Fe-4S]. Importantly, the reconstituted and as-isolated enzymes from yeast showed similar activities (~450 nmol ethylene min$^{-1}$ mg$^{-1}$), confirming that NifH$^{Ht}$ expressed in yeast mitochondria is provided with sufficient [Fe-S] clusters, which was not the case in rice. To enhance nitrogenase activity, it will therefore be necessary to characterize the insertion of [Fe-S] into NifH *in planta* and target this process for improvement.

In summary, we have demonstrated the expression of NifH$^{Ht}$ and NifM$^{Av}$ at the mRNA and protein levels in stably transformed rice callus and regenerated plants. We generated rice plants targeting nitrogenase Fe protein to the mitochondria. Fe protein purified from rice was stable, soluble, and capable of electron transfer to NifDK$^{Av}$, confirming the incorporation of endogenous rice mitochondrial [4Fe-4S] clusters, albeit at low levels. The stable accumulation of Fe protein in transgenic rice, a major staple crop, represents a critical step toward the expression of a complete functional Nif complex as required to achieve BNF in cereals, although the levels of active Fe protein obtained in this study are likely not yet sufficient for this goal. Further research should focus on increasing the occupancy and/or stability of NifH [4Fe-4S] clusters, wiring NifH to the cellular electron transfer machinery, and making the overall energetics more favorable for nitrogenase maturation and activity in plant mitochondria.

## Methods

**Genetic constructs.** The *H. thermophilus nifH* and *A. vinelandii nifM, nifS, nifU* genes were codon optimized for *S. cerevisiae* using the GeneOptimizer tool (Thermo Fisher Scientific, Waltham, MA, USA) and synthesized by Thermo Fisher Scientific as part of the Engineering Nitrogen Symbiosis for Africa (ENSA) project. The *ZmUbi1* + 1sti:cox4-twinstrep-*nifH*$^{Ht}$:tNos construct was transferred to vector pUC57 (GenScript, Piscataway, NJ, USA) by digesting the empty vector with BamHI and PstI and inserting the synthetic *ZmUbi1* + 1sti sequence, then digesting the intermediate vector with PstI and SphI and inserting the synthetic cox4-twinstrep-*nifH*$^{Ht}$:tNOS sequence. The pO*sActin*:su9-*nifM*$^{Av}$:tNos construct was transferred to pUC57 by digesting the empty vector with ACC65I and XbaI and inserting the *OsActin* sequence, then digesting the intermediate vector with XbaI and SalI and inserting the synthetic su9-*nifM*$^{Av}$:tNos sequence. The *ZmUbi1* + 1sti-su9-*nifS*$^{Av}$:tADH1 construct was transferred to vector pUC57 by digesting the empty vector with EcoRI and BamHI inserting the synthetic *ZmUbi1* + 1sti sequence, then digesting the intermediate vector with BamHI and HindIII and inserting the synthetic su9-*nifS*$^{Av}$:tADH1 sequence. The *ZmUbi1* + 1sti:su9-*nifU*$^{Av}$:tCYC1 construct was transferred to vector pUC57 by digesting the empty vector with EcoRI and SacI and inserting the synthetic *ZmUbi1* + 1sti sequence, then digesting the intermediate vector with SacI and BamHI and inserting the synthetic su9-*nifU*$^{Av}$:tCYC1 sequence. All restriction enzymes and T4 DNA ligase were obtained from Promega (Madison, WI, USA). Ligated plasmids were introduced into chemically competent *E. coli* DH5α cells and selected on lysogeny broth (LB) supplemented with 100 µg mL$^{-1}$ ampicillin. The inserts were confirmed by Sanger sequencing (Stabvida, Caparica, Portugal). The plant expression vectors, and the sequences of the genetic components, are listed in Supplementary Table 1. Primers used for vector construction are listed in Supplementary Table 2.

**Transformation of rice explants, callus recovery and regeneration of transgenic plants.** Six-day-old mature rice zygotic embryos (*Oryza sativa* cv. Nipponbare) were isolated from surface-sterilized seeds and transferred to Murashige and Skoog (MS) osmoticum medium prepared using 4.4 g L$^{-1}$ MS powder (Duchefa Biochemie, Haarlem, Netherlands) supplemented with 0.3 g L$^{-1}$ casein

hydrolysate, 0.5 g $L^{-1}$ proline, 72.8 g $L^{-1}$ mannitol, 30 g $L^{-1}$ sucrose and 2.5 mg L$^{-1}$ 2,4-dichlorophenoxyacetic acid (2,4-D) 4 h before bombardment with 10 mg gold particles coated with the *nifH* and *nifM* plasmids and the *hpt* plasmid at a 3:3:1 molar ratio[42]. The same method was used for co-transformation of *nifS* and *nifU* genes, to assess expression. The embryos were returned to osmoticum MS medium for 16 h before selection on standard MS medium (as above without mannitol) supplemented with 2.5 mg $L^{-1}$ 2,4-D and 50 mg $L^{-1}$ hygromycin in the dark for 2–3 weeks. One half of each callus clone was maintained in an undifferentiated state, and the other half was transferred to regeneration medium. Regenerated plantlets were transferred to soil and grown in large, flooded trays in growth chambers (28/25 °C day/night temperature with a 12-h photoperiod and 80% relative humidity)[43]. Plants were irrigated with tap water containing 100 μM Fe(III)-EDDHA (Sequestrene 138 Fe G-100; Syngenta Agro, Madrid, Spain). The Fe(III)-EDDHA solution in the trays was replaced every week. T0 plants were grown to an age of 1–2 months and harvested at the end of the dark period by cutting the whole plant down to 2–3 cm above soil level.

**Gene expression analysis by quantitative real-time PCR.** Total RNA was isolated from rice callus and corresponding regenerated plant leaves using the RNeasy Plant Mini Kit (Qiagen, Hilden, Germany). First-strand cDNA was synthesized from 1 μg total RNA using Ominiscript Reverse Transcriptase (Qiagen) and quantitative real-time PCR (qRT-PCR) was carried out by CFX96 system (Bio-Rad, Hercules, CA, USA) using a 15 μl mixture containing 1.0 μl 5-fold diluted cDNA template, 2× iTaq SYBR Green Supermix (Bio-Rad) and 0.5 μM of each primer[44]. The gene-specific primers listed in Supplementary Table 2. The identity of the PCR products was confirmed by sequencing. Expression levels were normalized against *OsActin* mRNA. Three technical replicates were tested for each sample.

**Protein extraction and immunoblot analysis.** Soluble rice protein extracts were prepared by grinding ~50 mg leaf tissue (snap-frozen in liquid $N_2$) in 2 mL Eppendorf tubes using 3 mm steel-balls and a BeadBug microtube homogenizer (Benchmark Scientific, Sayreville, NJ, USA) operating at 400 rpm for 20 s. Leaf powder was resuspended in seven volumes (v/v) of extraction buffer comprising 100 mM Tris-HCl (pH 8.6), 200 mM NaCl and 10% glycerol, supplemented with 1 mM PMSF, 1 μg $mL^{-1}$ leupeptin and 5 mM EDTA. After three rounds of homogenization, cell debris was removed by centrifugation at 20,000 × g for 5 min at 4 °C, and the supernatant was collected and stored at −80 °C.

Rice proteins were separated by SDS-PAGE and then immunoblotted to Protran Premium 0.45 μm nitrocellulose membranes (GE Healthcare, Chicago, IL, USA) using a semidry transfer apparatus (Bio-Rad) at 20 V for 45 min. Similar loading was confirmed by staining polyacrylamide gels with Coomassie Brilliant Blue or staining of nitrocellulose membranes with Ponceau S. The membranes were blocked with 5% non-fat milk in TBST (20 mM Tris-HCl pH 7.5, 150 mM NaCl, 0.02% Tween-20) for 1 h at room temperature before incubation with primary antibodies overnight at 4 °C. Primary monoclonal antibodies specific for strep-tagged *Os*NifH$^{Ht}$ (Strep-MAB, 2-1507-001; IBA Lifesciences, Göttingen, Germany), or polyclonal antibodies specific for NifH or NifM (polyclonal rabbit antibodies generated against *A. vinelandii* NifH and NifM proteins) or RuBisCO were diluted 1:2000–1:5000 in TBST supplemented with 5% bovine serum albumin (BSA). Secondary antibodies (Invitrogen, Thermo Fisher Scientific) were diluted 1:20,000 in TBST supplemented with 2% non-fat milk and incubated for 2 h at room temperature. Membranes were developed on medical X-ray films (AGFA, Mortsel, Belgium) using enhanced chemiluminescence.

**Purification of NifH by STAC.** *Os*NifH$^{Ht}$ protein extracts were prepared for STAC purification at $O_2$ levels below 1 ppm in anaerobic chambers (Coy Laboratory Products, Grass Lake, MI, USA, or MBraun, Garching, Germany). Callus (175 g for line HtH189, 234 g for HtH200 and 55 g for HtH202) was snap-frozen in liquid nitrogen, transferred inside the glovebox and then resuspended in lysis buffer comprising 100 mM Tris-HCl (pH 8.5), 200 mM NaCl, 10% glycerol, 3 mM sodium dithionite (DTH), 5 mM 2-mercaptoethanol (2-ME), 1 mM PMSF, 1 μg $mL^{-1}$ leupeptin, 10 μg $mL^{-1}$ DNAse I, and 1:200 (v/v) Bio-Lock solution (2-0205-050, IBA Lifesciences) at a ratio of 1:2 (w/w). For purification of *Os*NifH$^{Ht}$ from HtH200 plants, above-ground tissue (45 and 90 g for 1- and 2-months-old plants, respectively) from Fe fertilized plants was harvested at the end of the 12 h dark cycle (before the onset of light), snap-frozen in liquid nitrogen, transferred inside the glovebox and then resuspended in the lysis buffer at a ratio of 1:4 (w/v). The amounts of callus and plant tissue used for purifications are reported as fresh-weight. Callus and plant total extracts were prepared by mechanical disruption under an anaerobic atmosphere using a blender (Oster 4655) modified with a water-cooling system operating at full speed in four cycles of 2 min at 4 °C. The total extract was transferred to centrifuge tubes equipped with sealing closures (Beckman Coulter, Brea, CA, USA) and centrifuged at 50,000 × g for 1.5 h at 4 °C using an Avanti J-26 XP device (Beckman Coulter). The supernatant was passed through filtering cups with a pore size of 0.2 μm (Nalgene, Thermo Fisher Scientific) to produce a cell-free extract containing soluble proteins. This was loaded at a flow rate of 2.5 mL min$^{-1}$ onto a 5 mL Strep-Tactin XP column (IBA LifeSciences) attached to an ÄKTA FPLC system (GE Healthcare). The column was washed at 16 °C with

150 mL washing buffer comprising 100 mM Tris-HCl (pH 8.0), 200 mM NaCl, 10% glycerol, 2 mM DTH and 5 mM 2-ME. Bound proteins were eluted with 15–20 mL washing buffer supplemented with 50 mM biotin (IBA LifeSciences). The elution fraction was concentrated using an Amicon Ultra centrifugal filter with a cut-off pore size of 10 kDa (Merck Millipore, Burlington, MA, USA). Biotin was removed by passing the protein through PD-10 desalting columns (GE Healthcare) equilibrated with washing buffer. Desalted eluate was further concentrated and snap-frozen in cryovials (Nalgene) and stored in liquid nitrogen.

**Quantification of purified *Os*NifH$^{Ht}$ protein, N-terminal sequencing and MS analysis.** The purified *Os*NifH$^{Ht}$ protein was quantified by the densitometric analysis of Coomassie-stained gels using ImageJ[45] compared to a standard of yeast *Sc*NifB$^{Mi}$ protein as shown in Supplementary Fig. 4. N-terminal amino acid sequences were determined by Edman degradation (Centro de Investigaciones Biológicas, Madrid, Spain). *Os*NifH$^{Ht}$ protein (~250 pmol) was separated by SDS-PAGE, transferred to a Sequi-Blot PVDF membrane with a 0.2 μm pore size (Thermo Fisher Scientific) in 50 mM borate buffer (pH 9.0), stained with freshly prepared Coomassie R-250 (Sigma-Aldrich, St Louis, MO, USA) (0.1% in 40% methanol and 10% acetic acid), and then destained using 50% methanol. *Os*NifH$^{Ht}$ protein (~60 pmol) was separated by SDS-PAGE followed by Coomassie staining and destaining as described above. Protein excised from the gel was analyzed by mass spectrometry at the Universidad Complutense de Madrid, Spain.

**In vitro NifH activity analysis.** *Os*NifH$^{Ht}$ activity was determined in an anaerobic chamber as described[15]. Purified *Os*NifH$^{Ht}$ was analyzed using an acetylene reduction assay following the addition of NifDK$^{Av}$ and ATP-regenerating mixture (1.23 mM ATP, 18 mM phosphocreatine, 2.2 mM MgCl$_2$, 3 mM DTH, 46 μg $mL^{-1}$ creatine phosphokinase, 22 mM Tris-HCl pH 7.5) in a final volume of 400 μL in 9 mL serum vials containing 500 μL acetylene under an argon atmosphere. The assay was performed at 30 °C in a shaking water bath for 20 min. Reactions were stopped by adding 100 μL 8 M NaOH. Positive control reactions for acetylene reduction were carried out with NifH$^{Av}$. The ethylene formed in the reaction was measured in 50-μL gas-phase samples using a Porapak N 80/100 column in a gas chromatograph (Shimadzu, Duisburg, Germany). The ethylene peak from reactions devoid of both NifH and NifDK proteins (no nitrogenase component added) in each experiment was subtracted from the corresponding reported ARA activities.

**In vitro [Fe-S] cluster reconstitution and reconstituted NifH activity.** *Os*NifH$^{Ht}$ [Fe-S] cluster reconstitution in anaerobic chambers was carried out by incubating with *Ec*NifU$^{Av}$ (A. vinelandii NifU expressed in and purified from *E. coli* cells) previously loaded with [4Fe-4S] clusters. First, 20 μM NifU dimer was added to 100 mM MOPS (pH 7.5) supplemented with 8 mM 1,4-dithiothreitol (DTT) and the reaction was incubated at 37 °C for 30 min. We then added 1 mM L-cysteine, 1 mM DTT, 300 μM (NH$_4$)$_2$Fe(SO$_4$)$_2$, and 225 nM NifS$^{Av}$ purified from *E. coli* (*Ec*NifS$^{Av}$) to the reduced NifU and incubated the reaction on ice for 3 h. Finally, the proteins were diluted 40,000-fold in 100 mM MOPS (pH 7.5) and concentrated using Amicon centrifugal filters with a 30-kDa cut off to remove excess reagents. Isolated *Os*NifH$^{Ht}$ protein was mixed with the [4Fe-4S] cluster-reconstituted *Ec*NifU$^{Av}$ (NifU-mediated reconstitution) and then immediately used for the acetylene reduction assay.

**In vitro FeMo-co synthesis and apo-NifDK$^{Av}$ reconstitution.** NifB-dependent FeMo-co synthesis assays were prepared in anaerobic chambers[15]. We assembled 100 μL reactions containing 3 μM *Os*NifH$^{Ht}$, 1 μM *Sc*NifB$^{Mt}$, 9 μM [4Fe-4S] cluster-loaded *Ec*NifU$^{Av}$, 1.5 μM apo-NifEN$^{Av}$, 0.3 μM apo-NifDK$^{Av}$, 125 μM SAM, 17.5 μM Na$_2$MoO$_4$, 175 μM R-homocitrate, 1 mg $mL^{-1}$ BSA, and ATP-regenerating mixture (1.23 mM ATP, 18 mM phosphocreatine, 2.2 mM MgCl$_2$, 3 mM DTH, 46 μg $mL^{-1}$ creatine phosphokinase, 100 mM MOPS pH 7.5) at 30 °C for 1 h. In the positive control reaction, the assay was carried out by using *Sc*NifH$^{Ht}$ instead of the rice purified variant, and NifB-co (20.4 μM Fe) instead of *Sc*NifB$^{Mt}$ and *Ec*NifU$^{Av}$. In the negative control reaction, the reaction mixture contained all the components except for any source of NifB-co or *Sc*NifB$^{Mt}$ and *Ec*NifU$^{Av}$. Following the in vitro synthesis and insertion of FeMo-co, 17.5 μM (NH$_4$)$_2$MoS$_4$ was added to prevent further FeMo-co incorporation into apo-NifDK$^{Av}$, and the reaction was incubated for 20 min at 30 °C, shaking at 600 rpm. Activation of apo-NifDK$^{Av}$ was analyzed by adding 500 μL ATP-regenerating mixture and NifH$^{Av}$ (2.0 μM final concentration) in 9-mL vials containing 500 μL acetylene under argon gas. The acetylene reduction assay was performed at 30 °C for 20 min, and the resulting ethylene was measured in 50 μL gas-phase samples using a Porapak N 80/100 column as described above.

**Statistics and reproducibility.** For enzymatic activity assays in plants the standard deviation (SD) of the mean was calculated based on two biological replicates (two technical replicates each). The standard deviation (SD) of the mean for callus activity assays was calculated based on three biological replicates (three technical

replicates each). For relative gene expression analysis standard deviations (SD) were calculated based on three technical replicates.

## Data availability

The authors declare that the data supporting the findings of this study are available within the article, its Supplementary Information and data (Supplementary Data 1), and upon request.

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

## Acknowledgements

This work was supported, in whole or in part, by the Bill & Melinda Gates Foundation (OPP1143172 and INV-005889). Under the grant conditions of the foundation, a Creative Commons Attribution 4.0 Generic License has already been assigned to the Author Accepted Manuscript version that might arise from this submission. C.B. and W.H. were supported by doctoral fellowships from AGAUR. X.J. was supported by a doctoral fellowship from Universidad Politécnica de Madrid. This manuscript is dedicated to the memory of Dr. Changfu Zhu.

## Author contributions

C.B., S.B., X.J., L.M.R., and P.C., designed experiments. C.B., S.B., W.H., X.J., and T.C., performed the experiments and analyzed the data. C.B., S.B., L.M.R. and P.C. wrote the paper. All authors reviewed and approved the manuscript.

## Competing interests

The authors declare no competing interests.
