## [Peer Review File · Communications Biology]

Reviewers' comments:

Reviewer #1 (Remarks to the Author):

This manuscript outlines for the first time, to my knowledge, expression of functional component, FeProtein, of nitrogenase when expressed in stably transformed plants (rice as a major crop) where the components have been sent to the plant mitochondria. Building on an excellent body of prior research from these laboratories the authors show that expression of a variant NifH (possibly a thermotolerant variant) and NifM are capable of producing a partially functional FeProtein when the plant material is harvested at the end of the dark period (presumably to avoid oxygen produced during photosynthesis) and when the plants were fed supplementary Fe fertilizer. This reviewer appreciates the technical difficulties of the functional assays combining both plant-based extracts with various other components (isolated from bacterial and yeast) to general insights into the plant produced FeProtein. In this respect the manuscript outlines that the plant produced FeProtein as isolated (but under conditions outlined above) is capable of both donating electrons to the nitrogenase NifDK, and also the ability to contribute to FeMoco biogenesis. It is totally acceptable that these assays were conducted 'in vitro' with known components. Although it is not commented on in the manuscript it is highly likely that the involvement of the plant FeProtein in the second assay - FeMoco synthesis - is a two part reaction....where the first stage of the reaction is the reconstitution of the apo-FeProtein from the added-NifU, and from that starting point the holoFeProtein is capable of FeMoco synthesis. A significant outcome of this manuscript is that the loading of iron sulphur clusters into FeProtein is limiting for the observed activity. When the FeS clusters were repaired or added in vitro the activity assays increased significantly, demonstrably with the electron donation assay (ARA) and most likely implied for the FeMoco biogenesis assay (for reasons outlined above I can so no easy alternative method to avoid the inadvertent loading of iron sulphur clusters onto apo-FeProtein during that assay).

The authors spend a considerable amount of the discussion outlining the various reasons why iron sulphur clusters may not be loaded into FeProtein, and I think a significant improvement to the manuscript could be made by significantly expanding any results regarding the overexpression of AvNifS and AvNifU in the plants. Both NifS and NifU are the best candidates for loading iron sulphur clusters onto apoFeProtein and the authors state on line 256-258 that both of these proteins were attempted to be overexpressed in rice but somehow failed to generate detectable protein expression. The difficulty of detecting AvNifS or AvNifU could be placed in context of the difficulties these authors also found in expressing AvNifM (Figure 1). Indeed it's not entirely obvious whether or not NifM is absolutely required for FeProtein biogenesis- perhaps the authors have some other data to suggest that AvNifM is required for some aspect of NifH biogenesis in a plant expression context? Indeed relating to the requirement for NifM I am not happy that reference 21 is used to support the notion that NifM is required for folding NifH (see line 130). reference 21 contains no data referring to NifM, and there is controversial/contradictory evidence in the literature about the biochemical-molecular basis of NifM functionality.

So I will try to answer some of the points suggested for an incisive and well justified review.

1. Major claims of the paper are that a very oxygen sensitive metalloenzyme from bacterial nitrogenase can be stably expressed in a crop and that the crop suffers no apparent severe detriment impact. (Aspects of Supp figure 3a could be included in the main text to further strengthen that aspect). This particular component of nitrogenase performs two or three major functions in the overall biogenesis of nitrogenase and therefore result presented by these authors is of significant interest to those in the community engineering nitrogenase into crops. It is also of interest of those engineering oxygen sensitive pathways into crops and therefore I think it is of interest to a wider community.
2. To my knowledge stably transformed plants with FeProtein in the mitochondria is novel. By these same authors there is paper outlining the expression of NifH and NifM in stably-transformed plants where the NifHM are expressed in chloroplasts - and there is also suggestions of function in that paper (Aznar-Moreno et al., 2021 *Frontiers Agronomy*).

3. Strengthening the paper

- a. Again, like the prior paper Aznar-Moreno et al., 2021 *Frontiers Agronomy* (and also Jiang et al., 2021) and this submission there is not a clear series of experiments outlining the addition of iron supplementation nor the difference between harvesting in the light or harvesting in the dark.
 - b. I think the inclusion of negative data related to expression of NifM S and NifU is to be considered as an improvement. These are the obvious candidates for improving loading of apo-FeProtein and should be outlined further. Were the genes expressed that the mRNA level, not not detectable at the protein level etc? Are variant NifS and NifU to be considered, eg from HtNifS-U etc.
 - c. I see no evidence in that the literature regarding NifM beign required for folding of NifH proteins in plants. Prior papers always add AvNifM to the gene combinations and this is the same here. Is AvNifM needed for HtNifH to be active in isolates? Also reference 21 needs to be removed in the context used in Line 130.
 - d. The authors could also expand on the very complicated assays demonstrated in figure 4 and also how the as isolated plant (apo-) FeProtein is likely reconstituted during that FeMoco biogenesis assay.
 - e. Having gone to the trouble of producing stably-transformed plants and using promoters that are likely to be active in tissues other than leaves, the authors could leverage materials significantly by generating similar FeProtein extracts from other organs such as roots. Extraction of FeProtein from a non-leaf tissue would be an excellent reason for pursuing a crop expression system that is not considered 'transient expression' that can only be conducted in leaves.
4. Is the work convincing? Yes, the evidence surrounding the production of the GM crops is convincing as is the intricate biochemistry related to the functional assays as demonstrated. The statistics and graphics are convincing especially considering the complexity of the experiments undertaken.
5. Will the manuscript influence thinking in the field? To date numerous aspects of engineering nitrogenase into plants have relied upon so-called transient assays where gene combinations and pathways are investigated in a shapshot of expression a few short days after 'infiltration'. Whilst these assays have proven powerful for covering gene combinations and various molecular details, they can be criticised for being one step removed from being a genetically modified crop. In this instance this manuscript shows that stably transformed rice with a combination of NifH and NifM embedded in the nucleus and that the translated proteins sent to the plant mitochondria are capable of being isolated and found to be partially active. That these GM lines show no adverse phenotype is significant.

Pending all other reviews and advice, in my opinion this work can be published in this high impact journal if the points mentioned above are addressed.

Reviewer #2 (Remarks to the Author):

Expression of nifH as a functional protein in cereals is an important goal in establishing N₂fixation outside of legume. The groups who are involved in this work have made major progress in using *S. cerevisiae* as a model system to express the O₂ sensitive NifH protein in mitochondria. However, expression in plant mitochondria is an important next step and this is the first time this has been done without having to incubate plants or their extracts at very low O₂. The authors were able to show that an active NifH had been formed at low levels and activity and was only partially metalated with 4Fe-4S clusters. They were able to elegantly show that by reacting NifH from rice plants or callus with 4Fe-4S clusters in the added presence of NifU. It clearly shows that this step is likely to be the limiting step in achieving fully active NifH in rice plants. While there is very clearly lots of work to be done this is an important step in its own right.

Minor points

Line 71 I think there is one pair of P clusters per FeMo protein i.e. two P clusters per complex. Line 71 implies there are two pairs of 4 P clusters per FeMo protein.

Fig2 D. Probably my ignorance but I thought the twin Sterp tag (28 amino acids) ended QFEK, which is shown as part of NifH and not the yellow box for TwinStrep i.e. should the yellow box extend further?

Reviewer #3 (Remarks to the Author):

This work describes the construction of stable transgenic rice plants expressing functionally active NifH - the nitrogenase Fe-protein (dinitrogenase reductase) targeted to mitochondria. Two other proteins were also expressed in transgenic rice callus and plants: NifM - peptidyl prolyl cys-trans isomerase and hygromycin phosphotransferase (hpt gene). The level of expression of the fully mature NifH (dimeric NifH containing a [4Fe-4S] cluster) was of about 11% of the total expressed protein, since the apo-NifH could be activated in vitro by NifU and NifS by approximately 9fold. The NifH protein purified from rice plants and callus was capable of transferring electrons to nitrogenase (FeMo protein) and also to support the assembly/maturation of FeMo cofactor in the apo-NifDK, two essential/fundamental functions of NifH in nitrogen fixation. All experiments were well planned and all conclusions were supported by the results. Uncropped gels and immunoblots support all conclusions. The methodology is described in details and reproducible. The authors have supplied Mass Spectrometric information which confirm the identity of the expressed NifH protein.

The authors discuss the low level of maturation of the NifH protein in terms of low level of synthesis, insertion and or instability of the [4Fe-4S] cluster and present suggestions for future work.

All results were subjected to valid statistical analysis.

Protection of both the Fe-Protein and the MoFe-protein against inactivation by oxygen is a main issue in the regulation of nif genes expression and activity of the nitrogenase complex in bacteria. Two proteins involved in protection against O₂ are the Shethna protein in *Azotobacter vinelandii* and leghemoglobin in the nodule of rhizobia. these two should probably be considered in transgenic plants. This paper describes a breakthrough in the quest for transferring the capacity to fix nitrogen from prokaryotes especially to a very important crop, rice. This work is landmark in this endeavor.

A few questions:

1. How is the partition of apo-NifH between cytoplasm and mitochondria in transgenic rice callus and plants. In other word, were all NifH molecules transported into the mitochondria? If not, may this be the reason for the low activity of NifH as purified?
2. Is 10 mM O₂ low enough to prevent inactivation of NifH and NifD2K2 ?
3. In the three stable transgenic rice clones in which chromosome is the nifH construct inserted, since the three rice transgenic clones have different levels of NifH expression.
4. Have the authors attempted to increase the level NifH maturation in vivo in callus and rice plants by manipulating the level of oxygen?

Reviewer #1 (Remarks to the Author):

This manuscript outlines for the first time, to my knowledge, expression of functional component, FeProtein, of nitrogenase when expressed in stably transformed plants (rice as a major crop) where the components have been sent to the plant mitochondria. Building on an excellent body of prior research from these laboratories the authors show that expression of a variant NifH (possibly a thermotolerant variant) and NifM are capable of producing a partially functional FeProtein when the plant material is harvested at the end of the dark period (presumably to avoid oxygen produced during photosynthesis) and when the plants were fed supplementary Fe fertilizer. This reviewer appreciates the technical difficulties of the functional assays combining both plant-based extracts with various other components (isolated from bacterial and yeast) to general insights into the plant produced FeProtein. In this respect the manuscript outlines that the plant produced FeProtein as isolated (but under conditions outlined above) is capable of both donating electrons to the nitrogenase NifDK, and also the ability to contribute to FeMoco biogenesis. It is totally acceptable that these assays were conducted 'in vitro' with known components. Although it is not commented on in the manuscript it is highly likely that the involvement of the plant FeProtein in the second assay - FeMoco synthesis - is a two part reaction....where the first stage of the reaction is the reconstitution of the apo-FeProtein from the added-NifU, and from that starting point the holoFeProtein is capable of FeMoco synthesis. A significant outcome of this manuscript is that the loading of iron sulphur clusters into FeProtein is limiting for the observed activity. When the FeS clusters were repaired or added in vitro the activity assays increased significantly, demonstrably with the electron donation assay (ARA) and most likely implied for the FeMoco biogenesis assay (for reasons outlined above I can so no easy alternative method to avoid the inadvertent loading of iron sulphur clusters onto apo-FeProtein during that assay).

The authors spend a considerable amount of the discussion outlining the various reasons why iron sulphur clusters may not be loaded into FeProtein, and I think a significant improvement to the manuscript could be made by significantly expanding any results regarding the overexpression of AvNifS and AvNifU in the plants. Both NifS and NifU are the best candidates for loading iron sulphur clusters onto apoFeProtein and the authors state on line 256-258 that both of these proteins were attempted to be overexpressed in rice but somehow failed to generate detectable protein expression. The difficulty of detecting AvNifS or AvNifU could be placed in context of the difficulties these authors also found in expressing AvNifM (Figure 1). Indeed it's not entirely obvious whether or not NifM is absolutely required for FeProtein biogenesis- perhaps the authors have some other data to suggest that AvNifM is required for some aspect of NifH biogenesis in a plant expression context? Indeed relating to the requirement for NifM I am not happy that reference 21 is used to support the notion that NifM is required for folding NifH (see line 130). reference 21 contains no data referring to NifM, and there is controversial/contradictory evidence in the literature about the biochemical-molecular basis of NifM functionality.

So I will try to answer some of the points suggested for an incisive and well justified review.

1. Major claims of the paper are that a very oxygen sensitive metalloenzyme from bacterial nitrogenase can be stably expressed in a crop and that the crop suffers no apparent severe detriment impact. (Aspects of Supp figure 3a could be included in the main text to further strengthen that aspect). This particular component of nitrogenase performs two or three major functions in the overall biogenesis of nitrogenase and therefore result presented by these authors is of significant interest to those in the community

engineering nitrogenase into crops. It is also of interest of those engineering oxygen sensitive pathways into crops and therefore I think it is of interest to a wider community.

Response: Thank you for the suggestion. Supplementary Fig. 3 is now included in the main text as Fig. 1 E and F.

2. To my knowledge stably transformed plants with FeProtein in the mitochondria is novel. By these same authors there is paper outlining the expression of NifH and NifM in stably-transformed plants where the NifHM are expressed in chloroplasts - and there is also suggestions of function in that paper (Aznar-Moreno et al., 2021 *Frontiers Agronomy*).

Response: This is correct. The publication by Aznar-Moreno et al., 2021 *Frontiers Agronomy* is referred to in the manuscript (reference 23). In that study we show that NifH from *A. vinelandii* exhibited some activity when expressed together with NifM in tobacco chloroplasts, and when the NifH protein was isolated from leaves harvested at the end of the dark period.

3. Strengthening the paper

a. Again, like the prior paper Aznar-Moreno et al., 2021 *Frontiers Agronomy* (and also Jiang et al., 2021) and this submission there is not a clear series of experiments outlining the addition of iron supplementation nor the difference between harvesting in the light or harvesting in the dark.

Response: In the study by Aznar-Moreno et al. (2021, *Frontiers Agronomy*) we showed that AvNifH isolated from leaves exhibited significant activity only when harvested at the end of the dark period. In the study by Jiang et al. (2021, *Commun Biol*) we showed that HtNifH protein exhibited significant activity only when isolated from leaves of Fe fertilized tobacco at the end of an extended dark period (18 h instead of 8 h). Due to these previous observations, in the current study, we used the same protocol (Fe fertilization and harvesting of the rice tissue at the end of the dark period) to maximize chances of isolating active NifH from rice. This point is now highlighted in the Result and Methods sections of the revised manuscript.

b. I think the inclusion of negative data related to expression of NifM S and NifU is to be considered as an improvement. These are the obvious candidates for improving loading of apo-FeProtein and should be outlined further. Were the genes expressed that the mRNA level, not detectable at the protein level etc? Are variant NifS and NifU to be considered, eg from HtNifS-U etc.

Response: We felt this comment was important enough to carry out mRNA expression analysis for *nifS* and *nifU*. The results are now included in supplementary Figure 12. In summary, we measured high levels of *AvnifS* and *AvnifU* mRNA expressions in two independent rice lines (147 and 182), including regenerated plants from line 182. Repeated attempts to detect protein accumulation for NifS and NifU, did not result in detectable levels of these proteins. Our conclusion, therefore, is that the bottleneck in expression is not at the mRNA level. We have also discussed this in detail in our revised manuscript.

Supplementary Fig. 12 Relative mRNA expression (qRT-PCR) of *Azotobacter vinelandii nifS* and *nifU* in two independent rice callus lines (a) and regenerated plant line 182 (b). Callus line 147 failed to regenerate. Data (normalized to *OsActin* mRNA) are means \pm SD (n = 3 technical replicates). The sequences of the *nifS* and *nifU* genetic components are listed in Supplementary Table 1. Primers used for *nifS* and *nifU* vector construction and quantitative real-time PCR are listed in Supplementary Table 2.

c. I see no evidence in that the literature regarding NifM being required for folding of NifH proteins in plants. Prior papers always add AvNifM to the gene combinations and this is the same here. Is AvNifM needed for HtNifH to be active in isolates? Also reference 21 needs to be removed in the context used in Line 130.

Response: It is correct that reference 21 was inserted here inadvertently. It has now been replaced by the following references: Howard et al. (1986, JBC – for the expression of KpNifH in *E. coli*), Paya-Tormo et al. (2022, Sci Reports – for the expression of HtNifH and other NifH variants in yeast mitochondria) and Eserverri et al. (2020, PBJ – for the expression of AvNifH in tobacco chloroplasts). We are specifically referring to the recently published study where we show that AvNifM is required for HtNifH solubility in yeast mitochondria (Paya-Tormo et al., 2022). In that study, HtNifH expressed without AvNifM is mainly insoluble. However, the HtNifH protein that was isolated from these yeast cells (i.e. not expressing AvNifM) showed the same specific activity, but much lower total activity (activity per gram of yeast cells) as only a fraction of the HtNifH protein was soluble. Therefore, it appears that a HtNifH protein that is correctly folded (and hence accumulates as soluble protein) is indistinguishable from a HtNifH protein expressed together with AvNifM when it comes to specific activity. Therefore, in this manuscript we specifically write that “NifM has been shown to be required for NifH folding during Fe protein maturation” to avoid confusion between maturation and activity. We are open to rewiring this explanation if the referee feels it is still confusing.

d. The authors could also expand on the very complicated assays demonstrated in figure 4 and also how the as isolated plant (apo-) FeProtein is likely reconstituted during that FeMoco biogenesis assay.

Response: We have now expanded on the description of the FeMo-co synthesis assay. We have also highlighted that it is likely that the HtNifH protein from rice was reconstituted in the first step of the FeMo-co synthesis assay (when incubated with NifB and NifU).

e. Having gone to the trouble of producing stably-transformed plants and using promoters that are likely to be active in tissues other than leaves, the authors could leverage materials significantly by generating

similar FeProtein extracts from other organs such as roots. Extraction of FeProtein from a non-leaf tissue would be an excellent reason for pursuing a crop expression system that is not considered 'transient expression' that can only be conducted in leaves.

Response: Very good and sensible comment which we had also considered all along. In fact, we had tried to isolate the protein from rice roots growing hydroponically. It turned out that it was virtually impossible to accumulate enough biomass for these experiments as the total protein content in rice roots is very low. We list here some numbers to give a sense of the amounts required to isolate sufficient amount of recombinant protein for further experiments: The yield of NifH^{Ht} purified from *N. benthamiana* leaves in earlier reported studies was ~5–6 mg kg⁻¹. NifH^{Ht} from rice plants (0.25–0.5 mg kg⁻¹), the yield from rice callus (~6 mg kg⁻¹) was similar to that reported in *N. benthamiana* and the activity of the protein from rice callus was similar to that of the protein from *N. benthamiana* leaves.

4. Is the work convincing? Yes, the evidence surrounding the production of the GM crops is convincing as is the intricate biochemistry related to the functional assays as demonstrated. The statistics and graphics are convincing especially considering the complexity of the experiments undertaken.

5. Will the manuscript influence thinking in the field? To date numerous aspects of engineering nitrogenase into plants have relied upon so-called transient assays where gene combinations and pathways are investigated in a snapshot of expression a few short days after 'infiltration'. Whilst these assays have proven powerful for covering gene combinations and various molecular details, they can be criticized for being one step removed from being a genetically modified crop. In this instance this manuscript shows that stably transformed rice with a combination of NifH and NifM embedded in the nucleus and that the translated proteins sent to the plant mitochondria are capable of being isolated and found to be partially active. That these GM lines show no adverse phenotype is significant.

Pending all other reviews and advice, in my opinion this work can be published in this high impact journal if the points mentioned above are addressed.

Reviewer #2 (Remarks to the Author):

Expression of nifH as a functional protein in cereals is an important goal in establishing N₂ fixation outside of legume. The groups who are involved in this work have made major progress in using *S. cerevisiae* as a model system to express the O₂ sensitive NifH protein in mitochondria. However, expression in plant mitochondria is an important next step and this is the first time this has been done without having to incubate plants or their extracts at very low O₂. The authors were able to show that an active NifH had been formed at low levels and activity and was only partially metalated with 4Fe-4S clusters. They were able to elegantly show that by reacting NifH from rice plants or callus with 4Fe-4S clusters in the added presence of NifU.

It clearly shows that this step is likely to be the limiting step in achieving fully active NifH in rice plants. While there is very clearly lots of work to be done this is an important step in its own right.

Minor points

Line 71 I think there is one pair of P clusters per FeMo protein i.e. two P clusters per complex. Line 71 implies there are two pairs of 4 P clusters per FeMo protein.

Response: Corrected.

Fig2 D. Probably my ignorance but I thought the twin Strep tag (28 amino acids) ended QFEK, which is shown as part of NifH and not the yellow box for TwinStrep i.e. should the yellow box extend further?

Response: The boxed representation of the construct and the sequence are not to scale (please see the distance of the arrows in the two panels). It is correct that the TS-tag is 28 aa and ends with QFEK.

Reviewer #3 (Remarks to the Author):

This work describes the construction of stable transgenic rice plants expressing functionally active NifH - the nitrogenase Fe-protein (dinitrogenase reductase) targeted to mitochondria. Two other proteins were also expressed in transgenic rice callus and plants: NifM - peptidyl prolyl cys-trans isomerase and hygromycin phosphotransferase (hpt gene). The level of expression of the fully mature NifH (dimeric NifH containing a [4Fe-4S] cluster) was of about 11% of the total expressed protein, since the apo-NifH could be activated in vitro by NifU and NifS by approximately 9fold. The NifH protein purified from rice plants and callus was capable of transferring electrons to nitrogenase (FeMo protein) and also to support the assembly/maturation of FeMo cofactor in the apo-NifDK, two essential/fundamental functions of NifH in nitrogen fixation. All experiments were well planned and all conclusions were supported by the results. Uncropped gels and immunoblots support all conclusions. The methodology is described in details and reproducible. The authors have supplied Mass Spectrometric information which confirm the identity of the expressed NifH protein. The authors discuss the low level of maturation of the NifH protein in terms of low level of synthesis, insertion and or instability of the [4Fe-4S] cluster and present suggestions for future work. All results were subjected to valid statistical analysis. Protection of both the Fe-Protein and the MoFe-protein against inactivation by oxygen is a main issue in the regulation of nif genes expression and activity of the nitrogenase complex in bacteria. Two proteins involved in protection against O₂ are the Shethna protein in *Azotobacter vinelandii* and leghemoglobin in the nodule of rhizobia. These two should probably be considered in transgenic plants. This paper describes a breakthrough in the quest for transferring the capacity to fix nitrogen from prokaryotes especially to a very important crop, rice. This work is landmark in this endeavor.

A few questions:

1. How is the partition of apo-NifH between cytoplasm and mitochondria in transgenic rice callus and plants. In other word, were all NifH molecules transported into the mitochondria? If not, may this be the reason for the low activity of NifH as purified?

Response: As protein import is continuous during ongoing protein expression, there will always be a fraction of the protein pool in the cytoplasm (on the way of being imported into the mitochondria). Two lines of evidence indicate that the cytosolic apo-NifH fraction is small: first, N-terminal sequencing did not detect any unprocessed NifH protein; second, apo-NifH migrated mainly as a single band at the same position to the NifH protein targeted to yeast mitochondria. Therefore, the cytosolic NifH fraction does not seem to be the reason for the low activity of purified NifH.

2. Is 10 mM O₂ low enough to prevent inactivation of NifH and NifD2K2?

Response: The exact molarity of O₂ that will inactivate NifH and NifDK is unknown. However, we now know that the HtNifH protein is as sensitive, or even more O₂ sensitive, than AvNifH isolated from *A. vinelandii* (Paya-Tormo et al., 2022, Sci Reports).

3. In the three stable transgenic rice clones in which chromosome is the nifH construct inserted, since the three rice transgenic clones have different levels of NifH expression.

Response. As gene insertion is random into the rice genome, levels of expression in independent transformants will be different. This is a general rule for all transgenic plants irrespectively of how they are generated. A possible exception might be transplastomic plants which integrate input transgenes at defined locations in the plastid genome. Even in that case levels of expression among independent transformants do vary quite a bit. This variation is caused by many different factors, including the surrounding endogenous sequences, presence of cryptic enhancer elements in or close to the integration site, whether the input transgene has landed in an actively transcribing region, etc., etc. In fact the actual chromosome where the transgene has integrated matters very little as numerous studies have demonstrated already. In our experiments, the purified NifH protein was shown to be identical in the three independent lines and exhibited very similar activity.

4. Have the authors attempted to increase the level NifH maturation in vivo in callus and rice plants by manipulating the level of oxygen?

Response: No, we have not tried this. Instead of manipulating the external O₂ levels, in the future we would rather try to provide in vivo nitrogenase protection by modulating plant cell metabolism, in particular by controlling intracellular O₂ levels. In the recently published study of Paya-Tormo et al., 2022, Sci Reports, we identified a NifH variant that is more resistant to O₂ exposure. Finding Nif protein variants less sensitive to O₂, and the engineering of O₂ protection mechanisms within the plant cell, is one of the directions of our future research.

Reviewer #1 (Remarks to the Author):

I have reviewed the changes made by the authors, both in respect to their rebuttal letter and also the changes made to the manuscript.

In short all of my concerns and various suggestions for improvements and clarifications have been made. I thank the authors for the clear series of responses, the provision of extra data, modified wording where required, extra wording in methods and subtle nuance in some aspects of the discussion. The supplementary data is incredibly clear and I believe papers from these laboratories are setting a new standard for brevity in the main manuscript and also complete clarity in the associated supplementary data to ensure scientific rigour is adhered to. The protocols and experiments are incredibly intricate and complicated and the authors do an excellent service in sharing the methods used for this significant advance in the field of nitrogenase engineering into crops.

I only wording suggestion for the authors is to include the word "plant" in front of mitochondria on line 232 to make it clear that yeast mitochondria seem somehow to provide sufficient FeS clusters to NifH as opposed to plant mitochondria.

Reviewer #3 (Remarks to the Author):

This work describes the construction of stable transgenic rice plants expressing functionally active NifH - the nitrogenase Fe-protein (dinitrogenase reductase) targeted to mitochondria. Two other proteins were also expressed in transgenic rice callus and plants: NifM - peptidyl prolyl cys-trans isomerase and hygromycin phosphotransferase (hpt gene). The level of expression of the fully mature NifH (dimeric NifH containing a [4Fe-4S] cluster) was of about 11% of the total expressed protein, since the apo-NifH could be activated in vitro by NifU and NifS by approximately 9fold. The NifH protein purified from rice plants and callus was capable of transferring electrons to nitrogenase (FeMo protein) and also to support the assembly/maturation of FeMo cofactor in the apo-NifDK, two essential/fundamental functions of NifH in nitrogen fixation. All experiments were well planned and all conclusions were supported by the results. Uncropped gels and immunoblots support all conclusions. The methodology is described in details and reproducible. The authors have supplied Mass Spectrometric information which confirm the identity of the expressed NifH protein.

The authors discuss the low level of maturation of the NifH protein in terms of low level of synthesis, insertion and or instability of the [4Fe-4S] cluster and present suggestions for future work.

All results were subjected to valid statistical analysis.

Protection of both the Fe-Protein and the MoFe-protein against inactivation by oxygen is a main issue in the regulation of nif genes expression and activity of the nitrogenase complex in bacteria.

This paper describes a breakthrough in the quest for transferring the capacity to fix nitrogen from prokaryotes especially to a very important crop, rice.

The authors have accepted suggestions of the other reviewers, which have improved specific aspects of the manuscript.

All my queries have been convincingly answered.

Reviewer #1 (Remarks to the Author):

I have reviewed the changes made by the authors, both in respect to their rebuttal letter and also the changes made to the manuscript.

In short all of my concerns and various suggestions for improvements and clarifications have been made. I thank the authors for the clear series of responses, the provision of extra data, modified wording where required, extra wording in methods and subtle nuance in some aspects of the discussion. The supplementary data is incredibly clear and I believe papers from these laboratories are setting a new standard for brevity in the main manuscript and also complete clarity in the associated supplementary data to ensure scientific rigour is adhered to. The protocols and experiments are incredibly intricate and complicated and the authors do an excellent service in sharing the methods used for this significant advance in the field of nitrogenase engineering into crops.

I only wording suggestion for the authors is to include the word "plant" in front of mitochondria on line 232 to make it clear that yeast mitochondria seem somehow to provide sufficient FeS clusters to NifH as opposed to plant mitochondria.

Response: Thank you for the suggestion. The word “plant” has been included in front of “mitochondria”, on line 232, as suggested.

Reviewer #3 (Remarks to the Author):

This work describes the construction of stable transgenic rice plants expressing functionally active NifH - the nitrogenase Fe-protein (dinitrogenase reductase) targeted to mitochondria. Two other proteins were also expressed in transgenic rice callus and plants: NifM - peptidyl prolyl cys-trans isomerase and hygromycin phosphotransferase (hpt gene). The level of expression of the fully mature NifH (dimeric NifH containing a [4Fe-4S] cluster) was of about 11% of the total expressed protein, since the apo-NifH could be activated in vitro by NifU and NifS by approximately 9fold. The NifH protein purified from rice plants and callus was capable of transferring electrons to nitrogenase (FeMo protein) and also to support the assembly/maturation of FeMo cofactor in the apo-NifDK, two essential/fundamental functions of NifH in nitrogen fixation. All experiments were well planned and all conclusions were supported by the results. Uncropped gels and immunoblots support all conclusions. The methodology is described in details and reproducible. The authors have supplied Mass Spectrometric information which confirm the identity of the expressed NifH protein. The authors discuss the low level of maturation of the NifH protein in terms of low level of synthesis, insertion and or instability of the [4Fe-4S] cluster and present suggestions for future work.

All results were subjected to valid statistical analysis.

Protection of both the Fe-Protein and the MoFe-protein against inactivation by oxygen is a main issue in the regulation of nif genes expression and activity of the nitrogenase complex in bacteria.

This paper describes a breakthrough in the quest for transferring the capacity to fix nitrogen from prokaryotes especially to a very important crop, rice.

The authors have accepted suggestions of the other reviewers, which have improved specific aspects of the manuscript.

All my queries have been convincingly answered.